# Long-term impacts of an urban sanitation intervention on enteric pathogens in children in Maputo city, Mozambique: study protocol for a cross-sectional follow-up to the Maputo Sanitation (MapSan) trial 5 years postintervention

David A Holcomb [1,2] Vanessa Monteiro,[3] Drew Capone,[4] Virgílio António,[5] Márcia Chiluvane,[3] Victória Cumbane,[3] Nália Ismael,[5] Jackie Knee [6] Erin Kowalsky,[1] Amanda Lai [1] Yarrow Linden,[1] Elly Mataveia,[3] Rassul Nala,[7] Gouthami Rao,[1] Jorge Ribeiro,[3] Oliver Cumming [6] Edna Viegas,[3] Joe Brown [1]

**Correspondence to**
Dr Joe Brown;
joebrown@unc.edu

## ABSTRACT

**Introduction** We previously assessed the effect of an onsite sanitation intervention in informal neighbourhoods of urban Maputo, Mozambique on enteric pathogen detection in children after 2 years of follow-up (Maputo Sanitation (MapSan) trial, ClinicalTrials.gov: NCT02362932). We found significant reductions in *Shigella* and *Trichuris* prevalence but only among children born after the intervention was delivered. In this study, we assess the health impacts of the sanitation intervention after 5 years among children born into study households postintervention.

**Methods and analysis** We are conducting a cross-sectional household study of enteric pathogen detection in child stool and the environment at compounds (household clusters sharing sanitation and outdoor living space) that received the pour-flush toilet and septic tank intervention at least 5 years prior or meet the original criteria for trial control sites. We are enrolling at least 400 children (ages 29 days to 60 months) in each treatment arm. Our primary outcome is the prevalence of 22 bacterial, protozoan, and soil transmitted helminth enteric pathogens in child stool using the pooled prevalence ratio across the outcome set to assess the overall intervention effect. Secondary outcomes include the individual pathogen detection prevalence and gene copy density of 27 enteric pathogens (including viruses); mean height-for-age, weight-for-age, and weight-for-height z-scores; prevalence of stunting, underweight, and wasting; and the 7-day period prevalence of caregiver-reported diarrhoea. All analyses are adjusted for prespecified covariates and examined for effect measure modification by age. Environmental samples from study households and the public domain are assessed for pathogens and faecal indicators to explore environmental exposures and monitor disease transmission.

## STRENGTHS AND LIMITATIONS OF THIS STUDY

⇒ Long-term follow-up of an urban on-site sanitation intervention implemented at least 5 years prior so that all participants have been exposed to the treatment conditions for their entire lives.

⇒ Primary study endpoint is molecular detection of multiple enteric pathogens in child stool, which unambiguously indicates previous exposure to specific sanitation-related pathogens.

⇒ Primary outcome is the overall impact of the intervention on enteric pathogen exposure using a novel pooled estimate of the treatment effect across a prespecified set of enteric pathogens.

⇒ As an observational evaluation of an existing intervention, sample size is constrained by the number of eligible children residing at study sites and selection bias arising from differential enteric pathogen-related mortality may be present, particularly among older age groups.

⇒ As a cross-sectional study, there is potential for confounding bias in our estimates of the intervention impacts on child health, particularly as the intervention itself may have influenced the desirability of the intervention sites and thus the socioeconomic status of their residents, which may be associated with reduced pathogen exposures.

**Ethics and dissemination** Study protocols have been reviewed and approved by human subjects review boards at the Ministry of Health, Republic of Mozambique and the University of North Carolina at Chapel Hill. Deidentified study data will be deposited at https://osf.io/e7pvk/.
**Trial registration number** ISRCTN86084138.

## INTRODUCTION

There is a high burden of childhood enteric infections associated with poor environmental conditions, with multiple enteric pathogens frequently detected in stool within the first year of life.[1–3] This is also the period with the highest incidence of diarrhoea, which remains a leading cause of child mortality in low-income and middle-income countries and has long been associated with stunted growth.[4–6] However, many diarrhoeal episodes are not attributable to infectious aetiologies,[3 7] while the aetiology of attributable diarrhoea varies widely by setting and only certain diarrhoeal pathogens are consistently implicated in reduced linear growth.[2 8 9] Far more prevalent is asymptomatic enteric pathogen shedding in stool,[3 7 10–14] which may be more strongly associated with poor growth than diarrhoeal illness,[2 15–17] potentially by contributing to intestinal inflammation, gut permeability, and nutrient malabsorption in a condition known as environmental enteric dysfunction (EED).[18–21] In addition to the diverse negative impacts of stunting,[22] specific adverse health outcomes associated with enteric infection and EED include delayed cognitive development[23–25] and reduced oral vaccine efficacy.[26 27]

Water, sanitation and hygiene (WASH) interventions aim to prevent diarrhoea and improve child growth by interrupting faecal-oral pathogen transmission.[28] However, recent rigorous WASH intervention trials have demonstrated inconsistent and often limited impacts on child health.[13 29–34] Methodological limitations of the distal outcome measures used notwithstanding,[35 36] these findings are consistent with the interventions insufficiently interrupting environmental transmission and exposure to enteric pathogens.[37] Combined WASH interventions with high fidelity and adherence reduced stool-based detection of *Giardia*, hookworm and possibly *Ascaris* among 30-month-old children (but not younger children) and enteric viruses in rural Bangladesh,[10 38 39] *Ascaris* in rural Kenya,[40] and the number of codetected parasites in rural Zimbabwe,[11] but not bacterial pathogens or those most associated with stunting in any setting.[2] The sanitation-only intervention arm in rural Bangladesh reduced *Trichuris* prevalence in child stool[38] and may have reduced pathogenic *Escherichia coli* on child hands in more crowded households,[41] while a sanitation intervention in rural Cambodia did not impact any pathogens measured in child stool.[13] Viewed collectively, a clear picture emerges of pervasive childhood polymicrobial exposures that were not meaningfully prevented by low-cost WASH interventions.[42–44]

The preceding trials were all conducted in rural settings, but rapid urbanisation has led to exceptional growth of densely populated informal settlements that lack basic services and present unique health challenges.[45 46] We have previously investigated whether an onsite sanitation intervention delivered in low-income neighbourhoods of urban Maputo, Mozambique reduced enteric pathogen prevalence in child stool.[47] Although we similarly found no evidence of an effect on combined prevalence of prespecified enteric pathogens in our primary analyses,[34] additional evidence suggests that the sanitation intervention may have reduced exposures to enteric pathogens in the environment. The intervention was delivered with high fidelity and was widely used by intervention households after 2 years.[48] Exploratory subgroup analyses indicated that 24 months after the intervention, *Shigella* prevalence was halved (−51%; 95% CI −15% to −72%) and *Trichuris* prevalence reduced by three-quarters (−76%; 95% CI −40% to −90%) among children born into the study compounds after the intervention was implemented, relative to children born into the control compounds after baseline.[34] Furthermore, overall pathogen prevalence, pathogen counts, *E. coli* gene copy density, and the individual prevalence of *Ascaris* and pathogenic *E. coli* were all significantly reduced in soil at the intervention latrine entrance,[49 50] suggesting the intervention effectively contained human excreta. While animals have been implicated as major sources of pathogen exposure in other settings,[51] only companion animals were frequently present at study households and a locally validated indicator of poultry faecal contamination (the most commonly observed non-companion animal type) was rarely detected in household environments.[52] Conversely, indicators of human faecal contamination were widespread[50] and the two pathogens most impacted by the intervention—*Shigella* and *Trichuris*—are considered anthroponotic,[53] suggesting that human excreta was a primary source of enteric pathogens in this dense, urban setting and that human-associated pathogens were impacted by sustained exposure to the intervention.

Enteric pathogen carriage is highly age-dependent,[8 12] but we were previously limited by the timing of our follow-up survey (24 months postintervention) to children under 2 years of age when assessing impacts on those born into the intervention. Furthermore, it has been suggested that the typically short (1–2 years) follow-up periods often used in WASH evaluations may be inadequate for any longer-term benefits to manifest, potentially contributing to the limited observed effects.[42] Accordingly, we are conducting a cross-sectional follow-up study to better understand the long-term impacts of the sanitation intervention 5 years after it was implemented on child enteric pathogen exposures and shedding, diarrhoeal disease, and growth among children born into study households postintervention.

## STUDY OBJECTIVES

### Primary objective

To measure the long-term effect of a shared, onsite urban sanitation intervention on the pooled prevalence of prespecified enteric pathogen targets detected in children's stool at least 60 months postintervention.

### Hypotheses

H1: The risk of stool-based enteric pathogen detection among children 29 days to 60 months old is reduced for

children born into households that previously received the sanitation intervention.

H2: Children born into households that previously received the sanitation intervention experience delayed exposure to enteric pathogens relative to comparably aged children from non-intervention households, reflected in a greater reduction in the risk of enteric pathogen detection among younger age groups and attenuated reduction in risk among older children.

### Secondary objectives

1. To measure the effect of a shared, onsite urban sanitation intervention programme on the individual prevalence and density of prespecified enteric pathogen targets detected in children's stool and environmental samples at least 60 months postintervention.
2. To measure the effect of a shared, onsite urban sanitation intervention programme on child growth and the prevalence of caregiver-reported diarrhoea in children at least 60 months postintervention.

## METHODS AND ANALYSIS
### Study setting

This household-based study is being conducted in 11 bairros (neighbourhoods) in the Nhlamankulu district and 5 bairros in the KaMaxaquene district of Maputo city, Mozambique. Households in these densely populated, low-income neighbourhoods typically cluster into self-defined compounds delineated by a wall or boundary that share outdoor living space and sanitation facilities. Study activities with participating households are conducted primarily in the compound shared outdoor space by staff from the Centro de Investigação e Treino em Saúde da Polana Caniço (Polana Caniço Health Research and Training Centre; CISPOC) in Maputo, Mozambique, from which the study is implemented and managed.

### Study design
#### Urban sanitation intervention

The non-governmental organisation (NGO) water and sanitation for the urban poor (WSUP) implemented a sanitation intervention in 2015–2016 at approximately 300 compounds in informal urban neighbourhoods of Maputo under a larger programme led by the Water and Sanitation Programme of the World Bank.[47] In these compounds, ranging between three and 25 households each, existing shared, unhygienic latrines were replaced with pour-flush toilets and a septic tank with a soak-away pit for the liquid effluent. Two intervention designs employing the same sanitation technology were implemented, with approximately 50 communal sanitation blocks (CSBs) and 250 shared latrines (SLs) constructed across 11 neighbourhoods that exhibited diversity across density and other key characteristics (susceptibility to flooding, relative poverty, access to water and sanitation infrastructure).[12 52 54] SLs served compounds with fewer than 21 residents and provided a single cabin with the intervention toilet, while CSBs provided an additional cabin for every 20 compound residents and other amenities including rainwater harvesting, municipal water connections and storage, and washing, bathing and laundry facilities.[34 55] WSUP also constructed facilities to the same specifications at other compounds in the neighbourhoods prior to 2015 and continued to deliver these intervention designs in the years following 2016.

#### Original controlled before-and-after study

To evaluate the impact of the WSUP sanitation intervention on child health, we conducted the Maputo Sanitation (MapSan) trial, a controlled before-and-after study of enteric pathogen detection in child stool (ClinicalTrials.gov: NCT02362932).[47] WSUP selected intervention compounds using the following criteria: (1) residents shared sanitation in poor condition as determined by an engineer; (2) the compound was located in the predefined implementation neighbourhoods; (3) there were no fewer than 12 residents; (4) residents were willing to contribute financially to construction costs; (5) sufficient space was available for construction of the new facility; (6) the compound was accessible for transportation of construction materials and tank-emptying activities; (7) the compound had access to a legal piped water supply and (8) the groundwater level was deep enough for construction of a septic tank. Control compounds were selected according to criteria 1, 3, 4 and 7 from the 11 neighbourhoods where the intervention was implemented and 5 additional neighbourhoods with comparable characteristics.[12]

The MapSan trial recruited an open cohort at three time points: baseline (preintervention), 12 months postintervention, and 24 months postintervention; children were eligible to participate if older than 1 month at the time of enrolment and under 48 months at baseline. We enrolled intervention and control compounds concurrently to limit any differential effects of seasonality or other secular trends on the outcomes. We found no evidence that the sanitation intervention reduced the combined prevalence of 12 bacterial and protozoan enteric pathogens (the prespecified primary outcome), the individual prevalence of any single bacterial, protozoan, soil-transmitted helminth (STH), or viral pathogen, or the period prevalence of caregiver-reported diarrhoea 12 and 24 months postintervention.[34] However, exploratory analyses indicated the intervention may have been protective against bacterial and STH infections among the cohort who were born into the intervention and may have reduced the spread of some pathogens into latrine entrance soils.

#### Long-term cross-sectional follow-up study

We are revisiting MapSan trial compounds at least 5 years after the sanitation intervention to conduct a cross-sectional survey of children who were born after the intervention was implemented. Due to substantial population turnover,[34] both intervention and control compounds are being identified using previously collected geolocation

data and their locations confirmed by study staff during site identification visits. Any compounds identified during the course of these mapping efforts that have received the sanitation intervention at least 5 years prior (built by WSUP to the same specifications) or match the original enrolment criteria for MapSan control compounds are also invited to participate, irrespective of previous participation in the MapSan trial. As an existing intervention, neither staff nor participants can be blinded to intervention status. We anticipate enrolment to continue for approximately 1 year and are again enrolling intervention and control compounds concurrently to limit any differential effects of seasonality and other secular trends across the enrolment period.

### Patient and public involvement

Patients and/or the public were not involved in the design, conduct, reporting or dissemination plans of this research.

### Eligibility criteria

We attempt to enrol all eligible children in each compound that either previously participated in the MapSan trial, has previously received an intervention latrine, or meets the original enrolment criteria for MapSan control compounds.[34] Participant inclusion criteria include:

1. Child aged 29 days to 60 months old.
2. Born into and residing in a compound enrolled in the 5-year follow-up study; in intervention compounds, the child must have been born following the delivery of the sanitation intervention.
3. Has continuously resided in the study compound for the preceding 6 months, or since birth if under 6 months of age.
4. Has a parent or guardian who is able to understand and complete the written informed consent process and allow their child to participate.

Children are excluded if they have any caregiver-indicated medical condition or disability that precludes participation in the study.

### Participant enrolment

Enrolment is conducted by trained study staff in either Portuguese or Changana, according to the respondent's preference, with written materials provided in Portuguese. We first obtain verbal consent from the compound leader to approach households in the compound for enrolment. We then seek written, informed consent to participate from the parent or guardian of each eligible child. Participation is entirely voluntary; guardians may decline their child's participation for any reason and can withdraw their child at any point. We began enrolling participants on 28 March 2022.

### Household visits and procedures

After locating study compounds, the household of a participating child is visited twice, typically on consecutive days. On the first day, trained study staff conduct written consent procedures; administer compound, household, and child questionnaires; record child anthropometry measures; collect environmental samples; and request the child's caregiver to retain a sample of the child's stool. The household is revisited the following day to collect a stool sample from the child and complete environmental sample collection. If the stool sample is unavailable, the study staff coordinate an additional visit to retrieve the stool. In the event that 7 days pass since the initial visit without collection of a stool sample, a qualified nurse visits the child to obtain a rectal swab.

After collection of the stool sample, deworming is offered to all household members >1 year old who have not been dewormed in the past year, unless pregnant or breast feeding. Deworming consultation and medication provision is conducted by Ministry of Health staff following the national guidelines for deworming procedures. Deworming is offered in-kind to all household members and leverages the household interaction to provide an important public health service. Besides deworming, this study offers no direct benefit to children participating in this study. No incentives are provided to study participants, but we provide 50 meticais (approximately US$1) of mobile phone credit on the caregiver's preferred network for each child to compensate for the costs incurred in communicating with the study team to arrange household visits. Households from which we collect food and/or large-volume water samples are reimbursed 50 additional meticais to offset these expenses.

### Environmental sample collection

We are sampling environmental compartments at a randomly selected subset of 100 intervention and 100 control compounds to represent compound-level and household-level exposures.[37] At the entrance to the compound latrine we collect soil, flies, and a large volume air sample, as well as faecal sludge from the latrine or septic tank and any animal faeces observed in the shared outdoor space.[49 50 56 57] One household is randomly selected among those households with children enrolled in the child health study, from which we collect swabs of flooring at the household entrance, flies in cooking area, prepared child's food, stored drinking water, and water from the household's primary source.[52 58 59]

We are also collecting environmental samples from the public domain to conduct environmental surveillance of pathogens circulating in the community[56 60]; such samples are not linked to specific individuals in any way. Sample matrices are informed by the compartments on Maputo's Excreta Flow Diagram,[61] including wastewater, surface and open drain water, soils in the vicinity of solid and faecal waste disposal locations, and wastewater effluent from hospitals treating COVID-19 patients. We identify sample locations using satellite imagery in consultation with technicians responsible for sanitation and drainage in Maputo. Samples are collected with written permission from the municipal government on a weekly basis for matrices with a single available sampling location (eg, wastewater treatment plant influent) and twice at all

other locations (once in each the rainy and dry seasons) to prioritise the geographic distribution of sampling locations across Maputo city. We aim to collect approximately 100 samples of solids (soil, faecal sludge), 100 large-volume liquid samples (wastewater, surface water, and open drains),[62] and 100 passive samples of liquids.[60]

## Study outcomes

### Enteric pathogen detection in stool

Stool-based molecular detection is performed for 27 enteric pathogens commonly implicated in both symptomatic and asymptomatic childhood infections globally, including those identified at the Global Enteric Multicenter Study site in Manhiça, Mozambique.[12 36 56 63] Reverse-transcription quantitative PCR is conducted by custom TaqMan Array Card (TAC; Thermo Fisher, Carlsbad, California, USA) to simultaneously quantify genetic targets corresponding to 13 bacterial pathogens (*Aeromonas* spp, *Campylobacter jejuni/coli*, *Clostridioides difficile*, *E. coli* O157, enteroaggregative *E. coli* (EAEC), enteropathogenic *E. coli* (EPEC), enterotoxigenic *E. coli* (ETEC), Shiga toxin-producing *E. coli* (STEC), enteroinvasive *E. coli* (EIEC)/*Shigella* spp, *Helicobacter pylori*, *Plesiomonas shigelloides*, *Salmonella enterica*, *Vibrio cholerae*), 4 protozoan parasites (*Cryptosporidium* spp, *Cyclospora cayetanensis*, *Entamoeba histolytica*, *Giardia* spp), 5 STH (*Ascaris lumbricoides*, *Ancylostoma duodenale*, *Necator americanus*, *Strongyloides stercolaris*, *Trichuris trichiura*) and 5 enteric viruses (adenovirus 40/41, astrovirus, norovirus GI/GII, rotavirus, sapovirus).[64] We also include the respiratory virus SARS-CoV-2 on the custom TAC to support surveillance through faecal waste streams.[65 66]

Our primary outcome is the prevalence of a prespecified subset of 22 enteric pathogens, including *Shigella* spp and *T. trichiura*, the two pathogens most impacted at 24 months among children born after the intervention was delivered.[34] As in the original MapSan study, we exclude enteric viruses from the primary outcome due to greater potential for direct contact transmission, which is unlikely to be impacted by the intervention.[34 47 67] Because we anticipate a combined prevalence (detection of at least one pathogen in a given stool sample) near 100%, and the intervention could plausibly increase the prevalence of some pathogens while reducing others, we do not define a composite prevalence outcome.[34 68] Rather, we will estimate the effect of the intervention on the prevalence of each pathogen individually, along with a pooled estimate of the intervention effect across the outcome set to serve as a summary of intervention's impact on pathogen prevalence that is directly comparable to the individual effect estimates.[69 70] The pooled effect of a single intervention across a set of related outcomes may be thought of as the treatment effect on a generic enteric pathogen. The pathogens for which individual effects can be more precisely estimated (such as those with higher background prevalence) contribute more to the pooled estimate, which is recovered alongside the individual effects for each pathogen and shares the same scale,

allowing this summary metric to be directly interpreted in the context of its components, augmenting rather than replacing the individual estimates. This stands in contrast to composite outcomes, which construct new metrics that are related to, but fundamentally differ from, their component outcomes—the prevalence of *any* pathogen being conceptually distinct from the prevalence of a particular pathogen, for instance.

Secondary outcomes include the individual prevalence and the continuous gene copy density in stool of all 27 enteric pathogens assessed on TAC, including enteric viruses. Recognising the potential challenges to interpretation due to pervasive exposure, persistent infection and rapid reinfection, and possible protective effects,[71–74] we will also repeat the primary outcome analysis excluding *Giardia* from the outcome set.

### Anthropometry

Child weight and recumbent length (child age <24 months) or standing height (24–60 months) are assessed according to standard WHO protocols and transformed to age-adjusted z-scores using WHO reference populations to obtain height-for-age (HAZ), weight-for-age (WAZ), and weight-for-height (WHZ) z-scores.[75 76] Secondary outcomes include continuous HAZ, WAZ, and WHZ, as well as prevalence of binary growth outcomes stunting (HAZ <−2), underweight (WAZ <−2), and wasting (WHZ <−2).[13]

### Caregiver reported illness

Caregiver surveys are administered to ascertain child diarrhoeal disease, defined as the passage of three or more loose or watery stools in a 24-hour period, or any bloody stool, in the past 7 days.[77] We also assess two caregiver-reported negative control outcomes for each child: the 7-day period-prevalence of bruises, scrapes, or abrasions and the 7-day period-prevalence of toothache.[78] We do not expect the intervention to impact either child bruising or toothache prevalence, so significant differences in these outcomes by treatment arm would suggest possible bias in our caregiver-reported outcomes.

### Pathogen detection in environmental matrices

Molecular detection of selected pathogens and faecal source tracking (FST) markers is performed for environmental samples from both the private (compound and household) and public domains using a second custom TAC.[49 56 79 80] A subset of the enteric pathogens assessed in stool is included on the environmental TAC (adenovirus 40/41, astrovirus, norovirus GI/GII, rotavirus, sapovirus; *Aeromonas* spp, *C. jejuni/coli*, *C. difficile*, *E. coli* O157, EAEC; EPEC, ETEC, STEC, EIEC/*Shigella* spp, *H. pylori*, *S. enterica*, *V. cholerae*, *Cryptosporidium* spp, *E. histolytica*, *Giardia* spp, *A. lumbricoides*, *A. duodenale*, *N. americanus*, *T. trichiura*), as well as select environmental and zoonotic pathogens (*Leptospira* spp, *Toxocara* spp),[53 81 82] other human pathogens detectable in faeces (SARS-CoV-2, Zika virus, HIV proviral DNA, *Plasmodium* spp, *Mycobacterium*

*tuberculosis*),[83–86] FST markers (human, poultry, and canine mitochondrial DNA; avian 16S rRNA),[87 88] and general bacterial and anthropogenic pollution/antimicrobial resistance markers (bacterial 16S rRNA, class 1 integron-integrase gene *intl1*).[89 90] We also culture faecal indicator bacteria (total coliforms and *E. coli*) using the IDEXX Colilert-18 and Quanti-Tray 2000 system.[88 91] We conduct ongoing genomic surveillance of SARS-CoV-2 viral lineages in wastewater samples using amplicon-based Illumina next generation sequencing.[92]

### Statistical analysis

#### Enteric pathogen outcomes

We will use mixed effects models with varying slopes and intercepts to simultaneously estimate individual treatment effects for each pathogen and the weighted-average intervention effect across all pathogens included in the model. The modelling approach, including illustrative model specifications for both binary and continuous outcomes and interpretation of model parameters, is described in the prespecified data analysis plan (https://osf.io/e7pvk/). Briefly, the observed value (detection or gene copy density) of each pathogen for every child will be included as a separate response in the model design matrix and the intercept, intervention effect slope, and the slopes of other covariates will all be allowed to vary by pathogen. The intercept will also vary by child and compound to account for repeated measures of multiple pathogens per child and multiple children per compound. Each set of pathogen-varying effects (eg, the pathogen-specific intervention effect slopes) will be structured as arising from a population of parameters with shared mean and variance,[93 94] where the population-level mean corresponds to the weighted-average expected effect across all pathogens and the population-level variance indicates the extent to which the effect may differ by pathogen. We will also estimate covariances between the sets of pathogen-varying effects to account for dependencies, for example, if the effect of the intervention on a specific pathogen is greater when the background prevalence of that pathogen (represented by the pathogen-specific intercept) is also higher. By partially pooling information between pathogens, this approach provides adaptive shrinkage of the individual treatment effect estimates for each pathogen as well as an estimate of the generalised effect across pathogens.[50 93] Such partial pooling of effect estimates helps control the false discovery rate for individual outcomes, mitigating the need for post hoc multiple comparison adjustments.[69 70] As a sensitivity analysis, we will also fit separate models for each pathogen, applying the Benjamini-Hochberg procedure to control the false discovery rate.[95]

The mixed effects models will be specified as Bayesian hierarchical models with regularising hyperpriors (see data analysis plan for discussion of prior distributions) and sampled using Markov chain Monte Carlo approaches.[50 93 96] For binary enteric pathogen detection outcomes, the canonical logistic link function will be used

for computational stability and a Bayesian parametric g-formula algorithm will be applied to estimate marginal prevalence ratios (PR) and differences (PD) from the posterior predictive distribution.[97] Posterior predicted probabilities will be sampled for each pathogen assuming all participants received the intervention and again assuming none received the intervention, leaving all other covariate values unchanged, to obtain weighted averages over the distribution of confounders in the sample population, as in marginal standardisation.[98] The posterior predicted PR and PD distributions for each pathogen will be approximated by the ratio and difference, respectively, of the posterior predicted probability draws under the all-treated and none-treated scenarios. Differences in average gene copy density (scaled by pathogen-specific sample standard deviation to facilitate comparison across pathogens) will be estimated as the measure of effect using linear Bayesian hierarchical models for semicontinuous enteric pathogen quantity outcomes, with non-detects considered true zeros and censoring used to create a zero class (as in Tobit regression; refer to data analysis plan for implementation details).[96 99] As a sensitivity analysis, we will also implement the two-stage parametric g-formula approach of Rogawski McQuade *et al* to estimate differences in average quantity separately for each pathogen, applying the Benjamini-Hochberg procedure to account for multiple comparisons.[11] Effect estimates will be summarised using the mean of the posterior predictive distribution to represent the expected effect size and the central 95% probability interval to describe the range of effect sizes compatible with the data (the 95% CI). Parameters with 95% CIs that exclude the null will be considered significant, although the magnitude and uncertainty of parameter estimates will also be considered holistically in evaluating evidence for clinically meaningful effects.[100]

#### Growth and caregiver-reported outcomes

The effects of the intervention on mean HAZ, WAZ, and WHZ; the prevalence of stunting, underweight, and wasting; and the period-prevalence of caregiver-reported diarrhoea and negative control outcomes (bruising, scrapes, and abrasions; toothache) will be analysed separately as secondary outcomes using generalised estimating equations and robust standard errors with exchangeable correlation structure and clustering by compound (the level at which the sanitation intervention was delivered).[13 34 101] The estimated difference in age-adjusted z-scores by treatment assignment will be used as the measure of effect for continuous anthropometry outcomes. The PR will be estimated by modified Poisson regression for binary growth status and caregiver-reported outcomes. We will not adjust for multiple comparisons.[31 102]

#### Covariates and effect measure modification

As a cross-sectional study of an existing intervention, all analyses will be adjusted for a set of covariates selected a priori as potential confounders of the sanitation-enteric

pathogen shedding relationship.[103] The adjustment set will include child age and sex, caregiver's education, and household wealth index; additional covariates will be considered in exploratory adjusted analyses.[10 11 34 104] Records missing covariate data will be excluded from primary analyses. Analyses will be repeated with missing data imputed by multivariate imputation using chained equations (MICE) as a sensitivity analysis.[34 105] In addition to covariates, caregiver-reported outcomes and continuous growth outcomes (HAZ, WAZ, and WHZ) will be used for imputation; derived binary growth statuses (stunting, underweight, and wasting) will be recalculated for each observation after conducting imputation procedures. Missing pathogen outcomes will not be imputed, nor will pathogens be used to inform imputation of other variables. However, datasets containing imputed covariates will be joined to observed pathogen outcomes by unique child identifier for sensitivity analyses; the Bayesian hierarchical models will be fit to each imputed dataset and draws from the posterior predictive distributions for parameters and contrasts of interest will be concatenated to obtain pooled estimates across the multiple imputed datasets.[106] The specific enteric pathogens detected are expected to be strongly related to child age.[8 34] We will examine effect measure modification of the primary and secondary outcomes stratifying by age group (1–11 months, 12–23 months, and 24–60 months).[3 107]

### Independently upgraded controls

We anticipate some of the control compounds may have independently upgraded their sanitation facilities to conditions comparable to the intervention. Control compounds with sanitation facilities observed to possess cleanable, intact hardscape slabs; pour-flush or water-sealed toilets; a functional ventilation pipe; and a fixed superstructure with sturdy walls and a secure door that ensure privacy during use are considered to have independently upgraded to conditions comparable to the intervention.[55] Children living in control compounds with independently upgraded latrines are enrolled but will be excluded from the main analyses of the intervention effects. Two sets of subgroup analyses will instead be conducted that include all participants: one in which children in independently upgraded controls are considered as part of the control (non-intervention) arm and again considered as part of the intervention arm. We will compare parameter estimates from the three sets of analyses to investigate whether the sanitation improvements independently available in the study communities are comparable to the full sanitation intervention package assessed in the MapSan trial in terms of child health impacts.

Eligible children residing in any compound that has received a WSUP intervention latrine will be considered part of the intervention arm in primary analyses. We record the current conditions of intervention facilities but will not exclude or otherwise adjust for either upgraded or degraded sanitation facilities in intervention

compounds in order to evaluate the long-term impacts of the intervention following extended use.

### Minimum detectable effect size

The number of participants will be constrained by the number of compounds in the study neighbourhoods that have received the sanitation intervention or meet the eligibility requirements for MapSan control compounds, most of which were previously enrolled in the MapSan trial. At the 24-month follow-up, an average of 2.5 children per compound were enrolled from 408 compounds.[34] Compound-level intraclass correlation coefficients were generally less than 0.1 for individual pathogens, corresponding to cluster variances of ~0.05. We calculate the minimum detectable effect size (MDES) on individual pathogen prevalence with 80% power, 5% significance level and 0.05 compound cluster variance for a conservative scenario with 200 compounds per treatment arm and 2 children enrolled per compound (for 800 children total, 400 per arm), a moderate scenario of 220 compounds per treatment arm and 2.5 children enrolled per compound (550 children per arm, 1100 total), and a maximal scenario of 300 compounds per arm, 2.5 children per compound (750 children per arm, 1500 total).[108] Across all scenarios, a minimum baseline (untreated) prevalence of 6–8% is required to reach 80% power for even the largest theoretical effect (nearly 100% reduction). The minimum detectable relative reduction in prevalence decreases (ie, smaller effect sizes are more readily detected) as pathogen prevalence increases towards 100% (figure 1). The difference between the scenarios on the multiplicative scale is relatively minor, with the relative reduction MDES largely driven by pathogen baseline prevalence. A pathogen with baseline prevalence below 15% must have its prevalence halved (PR<0.5) in order to attain 80% power, while a 25% reduction is detectable with 80% power for baseline prevalence of 34%–46% under the maximal and conservative scenarios, respectively. We expect the simultaneous consideration of multiple pathogens to reduce the MDES for the primary outcome pooled intervention effect by effectively increasing the sample size. Because this pooled effect is dependent on the prevalence of each pathogen considered and the correlations between them, we will conduct simulation analyses to characterise plausible MDES ranges for the pooled primary outcome treatment effect.[68 108]

### Limitations

As an observational, cross-sectional evaluation of an existing intervention, this study faces a number of limitations that may impact the generalisability of our findings. By design, all of the study participants will have been born after the intervention was implemented in order to evaluate intervention impacts among individuals across a range of ages who have been exposed to intervention for their entire lives. However, there is potential for confounding bias in our estimates of the intervention impacts on child health, particularly concerning socioeconomic factors

## Minimum Detectable Relative Reduction in Prevalence

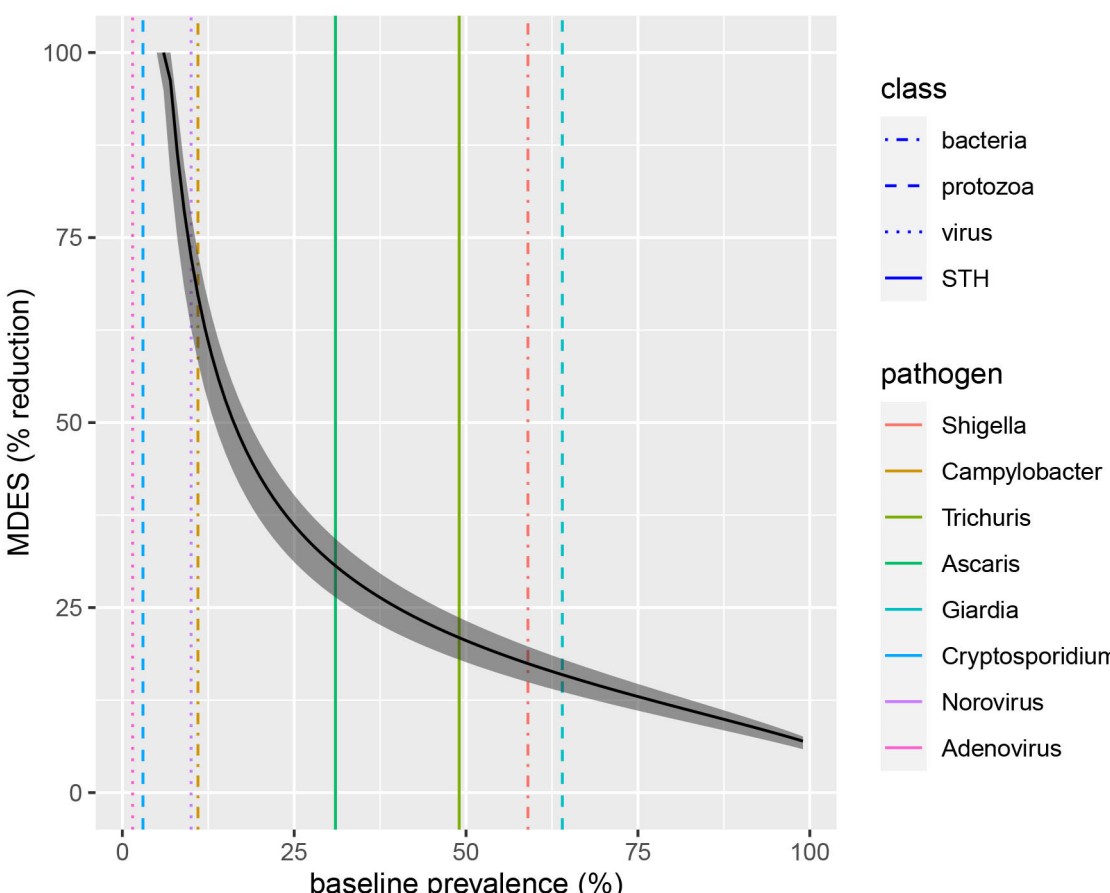

**Figure 1** Range of minimum detectable effect sizes (shaded region) for the per cent reduction in pathogen prevalence with 80% power, 5% significance level and 0.05 cluster variance across three sample size scenarios. The upper edge of the shaded area represents a conservative scenario with 800 total participants (2 per compound, 200 compounds per arm) while the lower edge corresponds to a maximal scenario with 1500 total participants (2.5 per compound, 300 compounds per arm). A moderate scenario with 1100 total participants (2.5 per compound, 220 compounds per arm) is represented by the black curve within the shaded area. The vertical lines show the prevalence of a subset of pathogens assessed in control compound children during the 24-month follow-up in the original MapSan trial. Line colour indicates the specific pathogen and line pattern reflects the pathogen class. MDES, minimum detectable effect size; STH, soil-transmitted helminth.

that may be associated with increased pathogen exposures. While non-randomised, the criteria for intervention and control compounds differed only by engineering considerations that we believe to be independent of the outcome. However, in the years since the intervention was implemented, the presence of the intervention itself may have influenced the desirability of the intervention sites and thus the socioeconomic status (SES) of their residents, particularly in light of the high population turnover previously observed after only 1–2 years. All analyses will be adjusted for a location-specific wealth index to account for potential differential SES between treatment arms,[104] in addition to other prespecified covariates associated with enteric pathogen exposure that may plausibly be related to treatment status, but the possibility of unmeasured and insufficiently controlled confounding remains.

Because childhood diarrhoea is a leading cause of child mortality, selection bias arising from survivor effects may

be present in our sample, particularly among older age groups.[109] However, loss to follow-up due to mortality was exceptionally rare in the previous assessment—far less common than emigration, which was not previously differential by treatment arm.[34] Our use of stool-based enteric pathogen detection as an objective primary outcome mitigates the potential for measurement bias[36] and we are assessing multiple negative control outcomes to account for potential response bias in our secondary caregiver-reported outcomes.[78] As a long-term evaluation of an existing intervention, the potential for exposure measurement error is low—interventions that have degraded substantially may shift results towards the null, but in so doing would represent a valid assessment of intervention sustainability.[110] The potential for exposure misclassification among controls is higher, in that they may have independently upgraded to sanitation infrastructure comparable to the intervention. We are actively monitoring this possibility at all study sites; have

prespecified criteria for identifying sites achieving such conditions; and will conduct sensitivity analyses with these sites excluded, considered as controls, and considered as equivalent to the intervention to characterise the impact of any such potential exposure misclassification among controls.

A key challenge is identifying an appropriate set of outcomes against which to evaluate the intervention; in this regard, we preferred inclusivity given the pathogens likely to be observed in our study setting, at the risk of unduly shifting our eventual results towards the null. This challenge persists, regardless of study design, so long as the study aims include assessing the effect of some condition or intervention on exposure to, or infection by, multiple pathogens that share a common route of transmission.[111]

## Contribution

Although observational, this study is unique in evaluating sanitation intervention effects at least 5 years after the intervention was implemented and up to 5 years after participants were borne into the intervention conditions. Previous studies have focused on WASH intervention impacts up to 2–3 years after delivery,[13 31–34] which may be insufficient time to realise impacts.[42] Furthermore, most previous studies have been conducted in rural settings, while we are investigating the long-term effects of an urban sanitation intervention that is broadly representative of the types of infrastructure improvements likely to be available in rapidly growing urban informal settlements in coming years. Finally, we use stool-based detection of multiple enteric pathogens as an objective outcome and propose a novel pooled estimate of the treatment effect across a prespecified outcome set to summarise the overall impact of the intervention on enteric pathogen exposure as the primary trial outcome.

## Ethics and dissemination

This study was approved by Comité Nacional de Bioética para a Saúde, Ministério da Saúde de Moçambique (FWA#: 00003139, IRB00002657, 326/CNBS/21; approved: 15 June 2021) and University of North Carolina at Chapel Hill Ethics Committee (IRB#: 21-1119; approved 19 August 2021) and was prospectively registered with ISRCTN on 16 March 2022 (https://doi.org/10.1186/ISRCTN86084138). We prespecified a statistical analysis plan that was deposited in a permanent online repository (https://osf.io/e7pvk/) prior to commencing enrolment. All protocol modifications will be submitted to, and approved by, the respective ethics committees prior to implementation and promptly updated on the ISRCTN registry.

Written, informed consent is obtained from the parent(s)/guardian(s) of each participant. Identifiable information is maintained separately to protect confidentiality and linked to surveys, stool, and environmental sample data by unique codes; the linking file to identifiable information will be destroyed on completion of the study and the study data and remaining biospecimens retained for fully deidentified ancillary analyses. Results will be presented to key stakeholders in Mozambique, including local and national government officials, public utilities, and NGOs, and published in open access peer-reviewed journals. On publication of study results, the underlying individual participant data will be fully deidentified and made freely available in the permanent online repository (https://osf.io/e7pvk/).

**Author affiliations**
[1]Department of Environmental Sciences and Engineering, Gillings School of Global Public Health, University of North Carolina, Chapel Hill, North Carolina, USA
[2]Department of Epidemiology, Gillings School of Global Public Health, University of North Carolina, Chapel Hill, North Carolina, USA
[3]Centro de Investigação e Treino em Saúde da Polana Caniço, Instituto Nacional de Saúde, Maputo, Mozambique
[4]Department of Environmental and Occupational Health, School of Public Health, Indiana University, Bloomington, Indiana, USA
[5]Division of Biotechnology and Genetics, Instituto Nacional de Saúde, Marracuene, Mozambique
[6]Department of Disease Control, Faculty of Infectious and Tropical Diseases, London School of Hygiene and Tropical Medicine, London, UK
[7]Division of Parasitology, Instituto Nacional de Saúde, Maputo, Mozambique

**Acknowledgements** We thank all the participants, their families and neighbours for graciously welcoming us into their communities, and our implementing partner, water and sanitation for the urban poor, for their continued support. We also gratefully acknowledge the hard work of the CISPOC survey team, including Jorge Binguane, Noémia Come, Anelsa Dunhe, Alice Fumo, Antônio Johane, Evelin Matos, Eloisa Mula, Mariza Rachid and Líria Sambo; and of Filipe Fazenda, Cláudia Machume, and Alfredo Muchanga in the CISPOC laboratory.

**Contributors** JB, EV, OC, RN and JK designed the study, secured funding and provided ongoing supervision. JB is also the corresponding author and guarantor. VM and DC drafted the initial study protocol and obtained ethical approvals. DAH developed the statistical analysis plan, prepared the prospective trial registration and protocol manuscript, and is the first author. EK, GR, EM, JR, VA, AL and YL developed and implemented the study procedures, which were overseen by VC, VM, MC and NI. All authors contributed to and approved the final manuscript.

**Funding** This work is funded by the Bill & Melinda Gates Foundation (OPP1137224) with additional support from a National Institute of Environmental Health Sciences training grant (T32ES007018).

**Disclaimer** The funders had no role in the study design; data collection, analysis, and interpretation; or decision to publish.

**Competing interests** None declared.

**Patient and public involvement** Patients and/or the public were not involved in the design, or conduct, or reporting, or dissemination plans of this research.

**Patient consent for publication** Not applicable.

**Provenance and peer review** Not commissioned; externally peer reviewed.

**ORCID iDs**
David A Holcomb http://orcid.org/0000-0003-4055-7164
Jackie Knee http://orcid.org/0000-0002-0834-8488
Amanda Lai http://orcid.org/0000-0002-3768-7294
Oliver Cumming http://orcid.org/0000-0002-5074-8709
Joe Brown http://orcid.org/0000-0002-5200-4148

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
