## [Reviewer comments · BMJ Open]

ARTICLE DETAILS

TITLE (PROVISIONAL)	Long-term impacts of an urban sanitation intervention on enteric pathogens in children in Maputo city, Mozambique: study protocol for a cross-sectional follow-up to the Maputo Sanitation (MapSan) trial five years post-intervention
AUTHORS	Holcomb, David; Monteiro, Vanessa; Capone, Drew; António, Virgílio; Chiluvane, Márcia; Cumbane, Victória; Ismael, Nália; Knee, Jackie; Kowalsky, Erin; Lai, Amanda; Linden, Yarrow; Mataveia, Elly; Nala, Rassul; Rao, Gouthami; Ribeiro, Jorge; Cumming, Oliver; Viegas, Edna; Brown, Joe

VERSION 1 – REVIEW

REVIEWER	Arnold, Ben UCSF, F.I. Proctor Foundation
REVIEW RETURNED	05-Oct-2022

GENERAL COMMENTS	General comments The study protocol describes an important, longer-term follow-up of the MapSan trial in Maputo, Mozambique. The original study has been one of the most important sanitation trials completed over the past 20 years, and this additional follow-up survey will be an invaluable, additional contribution to the broader WASH field. I had only a few comments related to the proposed primary outcome and statistical analysis, which I hope are constructive and lead to further clarification and transparency. I commend the authors on this important study. Ben Arnold, UCSF Major comments (1) Page 18 lines 53-55, Page 19 lines 3-10 “Because we anticipate a combined prevalence (detection of at least one pathogen in a given stool sample) near 100%, and the intervention could plausibly increase the prevalence of some pathogens while reducing others, we do not define a composite prevalence outcome [34,67]. Rather, we use mixed effects models to analyze all pathogens concurrently, thereby estimating the pooled effect of the intervention on enteric pathogen prevalence across the outcome set [68,69].” and Page 21, lines 25-33 “We will use mixed effects models with varying slopes and intercepts to estimate the weighted- average intervention effect across all pathogens included in the model, in essence treating
--

each pathogen as a separate study of the intervention effect on enteric pathogen detection. Additional random effects will be included to account for clustering by compound and child.”

(1a) In the outcomes section, the authors express their concern with composite outcomes across pathogens in a context where the intervention could affect only a small number of pathogens (as in the original endpoints at 24 months) or have effects in opposite directions depending on the pathogen. In either case, an average effect weighted across pathogens might show no effect and could potentially obscure important, pathogen-specific effects. Yet, the proposed analysis approach seems to target a “weighted average intervention effect” across 22 pathogens. This description of the statistical analysis approach seems at-odds with its goal. Please clarify how the proposed approach of a weighted-average effect across 22 pathogens somehow skirts the issues the authors describe with a composite outcome in this context.

(1b) The current description of the primary statistical analysis is vague and was difficult for me to understand. It might be a very reasonable approach, but it was difficult for me to tell. The authors propose to “use mixed effects models to analyze all pathogens concurrently” (p19, line 6), yet it is unclear to me how the described modeling approach would be truly multivariate. Are the authors proposing to model all 22 pathogen outcomes jointly? How many parameters would the model have? It would seem enormous, and potentially very difficult to fit given the number of children proposed in the study. Or, do the authors have in mind something akin to a 2-stage approach where the intervention effect is estimated separately for each of 22 pathogens (stage 1), and then those 22 separate estimates are then pooled in some manner (stage 2)? Please clarify, as the analysis approach is not commonly used (to my knowledge) for multiple outcome comparisons in trials.

(1c) Finally, it would be helpful to understand why a weighted average effect across 22 pathogens would be more interpretable than pathogen-specific estimates, or class-specific estimates (bacteria, protozoa, helminths, viruses). For example, what if there are 20 pathogens with clearly null effects and 2 pathogens with moderate reductions that appear to be real (as in the original analysis)? In this hypothetical scenario, wouldn’t a weighted average effect across the 22 show no effect? Would the interpretation at the end of the day somehow be different than if the trial simply reported 22 separate effects? If the decision to estimate a weighted average effect is driven by statistical power, I would expect that 22 separate estimates with a reasonable false discovery rate correction for multiple comparisons would provide nearly the same power and would be easier to understand/replicate and more interpretable.

(2)

Page 22, lines 54-56

“Analyses will be repeated with missing covariate data imputed by multivariate imputation using chained equations (MICE) as a sensitivity analysis [34, 99].”

This relates to my previous comment about vague descriptions in the statistical analysis, but please explain how the 22-outcome, multivariate model for the primary analysis would then be incorporated into a MICE routine for multiple imputation. It’s difficult

	for me to understand how this would work, and the citations provided do not illustrate how it would be done. Minor comments (3) Page 5, Lines 49-54. "...in particular, the presence of the intervention itself may have influenced the desirability of the intervention sites and thus the socioeconomic status of their residents, which may be associated with increased pathogen exposures" (also page 26, lines 6-41) Please confirm the sentence should have "increased pathogen exposures" rather than "reduced pathogen exposures." Intuitively, I would have thought that the intervention would have potentially increased the desirability of the intervention sites, increased the socioeconomic status of residents, and in turn reduced pathogen exposure. If it is the other way around, it was not intuitive to me. Either way, perhaps it would help to connect the dots a bit more for readers and provide the rationale. (4) On the relative scale, the analysis proposes to use an odds ratio as a measure of effect to compare the weighted average prevalence of infection. Since infection prevalence will be high for many pathogens, the odds ratio will not approximate the prevalence ratio and might be exaggerated in the extreme. For example, if prevalence in the two groups were 80% and 85%, then the prevalence ratio is $0.80/0.85 = 0.94$ (a 6% relative reduction), but the odds ratio is $(0.8/0.2)/(0.85/0.15) = 0.71$ (a 29% relative reduction) — quite a difference! Since the authors propose to estimate prevalence differences using marginal predictions, why not do something similar for a more interpretable prevalence ratio on the relative scale? Furthermore, the authors propose estimating a prevalence ratio for growth-related binary outcomes. It would seem like a consistent analysis approach and parameter of interest might be valuable across similar outcomes in the same study, or at least some justification for why they should differ. (5) Page 21, lines 42-46 "Prevalence differences (PD) between children in intervention and control compounds will also be estimated from the posterior predictive distribution at representative values of other model covariates [92]." Why a predictive distribution at fixed values rather than simply use marginal standardization? There are substantive differences in interpretation, and in general, marginal predictions are often easier to interpret: https://pubmed.ncbi.nlm.nih.gov/24603316/
--	---

REVIEWER	Khabo-Mmekoa, Colette Mmapenya Tshwane University of Technology, Biomedical Sciences
REVIEW RETURNED	19-Oct-2022

GENERAL COMMENTS	I have had a great opportunity to review the manuscript entitled "Long-term impacts of an urban sanitation intervention on enteric pathogens in children in Maputo city, Mozambique: study protocol for a cross-sectional follow-up to the Maputo Sanitation (MapSan) trial five years post-intervention", which is considered for
--

	publication in BMJ open. This work focuses on the health impacts of sanitation interventions after five years among children born into the study households post-intervention. The study, in my opinion, is well written, and the evaluations conducted aided in implementing better intervention conditions. Can the authors please specify when this study began? As stated on page 16, first paragraph, last sentence, participant enrollment did not begin until March 2022. When did the onsite sanitation interventions take place? Authors must take note of the citation of the references. All references are recorded accordingly up to page 22. Reference 100 is listed but not cited within the text. This may be a need for adjustment for the remaining references.
--	---

REVIEWER	Foster, Tim University of Technology Sydney
REVIEW RETURNED	11-Dec-2022

GENERAL COMMENTS	The paper details a comprehensive and balanced study protocol that builds on previous research in the study site. As currently designed, the study will provide important insights into the longer term outcomes associated with a sanitation intervention in urban Mozambique. The methods are clearly and transparently explained, as are the study's limitations. My only suggestion is in relation to the 'public domain' environmental sampling, where a bit more detail might be helpful. While the protocol articulates the approach for the environmental sampling at control and intervention compounds, I could not find information on the proposed sample size for the public domain sampling, the expected distribution across public domain sample types (wastewater, surface water, soil etc) or how the public domain locations would be selected.
---

VERSION 1 – AUTHOR RESPONSE

Reviewer 1:
Dr. Ben Arnold, UCSF

(1a)

In the outcomes section, the authors express their concern with composite outcomes across pathogens in a context where the intervention could affect only a small number of pathogens (as in the original endpoints at 24 months) or have effects in opposite directions depending on the pathogen. In either case, an average effect weighted across pathogens might show no effect and could potentially obscure important, pathogen-specific effects. Yet, the proposed analysis approach seems to target a “weighted average intervention effect” across 22 pathogens. This description of the statistical analysis approach seems at-odds with its goal. Please clarify how the proposed approach of a weighted-average effect across 22 pathogens somehow skirts the issues the authors describe with a composite outcome in this context.

This is an excellent point, and we certainly do not intend to assert that our proposed weighted-average intervention effect (aka “pooled intervention effect”) fully addresses the issue of summarizing heterogenous intervention effects across multiple outcomes in a single metric. Our approach is

motivated by analogy to meta-analysis, in which the (potentially heterogenous) effects estimated under multiple studies are pooled to obtain a weighted average effect across the studies. We might consider each specific pathogen assessed as a separate study of the effect of the intervention on enteric pathogens generally; by “meta-analyzing” the pathogen-specific intervention effects, we obtain a pooled estimate that may be interpreted as the expected effect of the intervention on a generic enteric pathogen. Such pathogen-specific “studies” are obviously related (as they are measured in the same stools) and should not be treated as independent, but pooling the effects across pathogens provides a useful summary of the overall impact of the intervention, considering sanitation’s role in controlling all pathogens (ideally) originating in human excreta. Crucially, this summary may be directly compared to and interpreted within the context of the individual treatment effects for specific pathogens. The same does not hold for composite outcomes, which define new metrics that, while potentially on the same scale (e.g., prevalence of any pathogen and prevalence of a specific pathogen), are nevertheless conceptually distinct from their constituent outcomes. However, while we believe this approach is an improvement over composite outcomes, all the same challenges and pitfalls attendant to meta-analysis would apply to our proposed pooled effect estimate, not least of which the proverbial “garbage-in, garbage-out”, which is particularly relevant to defining the appropriate outcome set over which to estimate the pooled treatment effect. While space constraints do not allow a full treatment of this subject (which we intend to address in future work that also characterizes the power and bias of the proposed approach using simulation studies informed by real-world data), we have revised the text in several ways to clarify our proposed approach. First and foremost, we have updated the primary outcome definition to include the treatment effects on the prevalence of each specific pathogen so that the pooled effect might be interpreted in the appropriate context (lines 356-367):

Our primary outcome is the prevalence of a pre-specified subset of 22 enteric pathogens, including *Shigella* spp. and *T. trichiura*, the two pathogens most impacted at 24 months among children born after the intervention was delivered [34]. As in the original MapSan study, we exclude enteric viruses from the primary outcome due to greater potential for direct contact transmission, which is unlikely to be impacted by the intervention [34,47,67]. Because we anticipate a combined prevalence (detection of at least one pathogen in a given stool sample) near 100%, and the intervention could plausibly increase the prevalence of some pathogens while reducing others, we do not define a composite prevalence outcome [34,68]. Rather, we will estimate the effect of the intervention on the prevalence of each pathogen individually, along with a pooled estimate of the intervention effect across the outcome set to serve as a summary of intervention’s impact on pathogen prevalence that is directly comparable to the individual effect estimates [69,70].

This will ensure that any important effects on specific pathogens will not be overlooked in the event that the overall average effect is null; instead, the pooled effect is presented as a complement to the treatment effects on individual pathogens. We have also expanded the rationale for estimating a pooled outcome across all pathogens (lines 367-381):

The pooled effect of a single intervention across a set of related outcomes may be thought of as the treatment effect on a generic enteric pathogen. The pathogens for which individual effects can be more precisely estimated (such as those with higher background prevalence) contribute more to the pooled estimate, which is recovered alongside the individual effects for each pathogen and shares the same scale, allowing this summary metric to be directly interpreted in the context of its components, augmenting rather than replacing the individual estimates. This stands in contrast to composite outcomes, which construct new metrics that are related to, but fundamentally differ from, their component outcomes—the prevalence of any pathogen being conceptually distinct from the prevalence of a particular pathogen, for instance.

(1b)

The current description of the primary statistical analysis is vague and was difficult for me to understand. It might be a very reasonable approach, but it was difficult for me to tell. The authors propose to “use mixed effects models to analyze all pathogens concurrently” (p19, line 6), yet it is unclear to me how the described modeling approach would be truly multivariate. Are the authors proposing to model all 22 pathogen outcomes jointly? How many parameters would the model have? It would seem enormous, and potentially very difficult to fit given the number of children proposed in the study. Or, do the authors have in mind something akin to a 2-stage approach where the intervention effect is estimated separately for each of 22 pathogens (stage 1), and then those 22 separate estimates are then pooled in some manner (stage 2)? Please clarify, as the analysis approach is not commonly used (to my knowledge) for multiple outcome comparisons in trials.

We appreciate the opportunity to clarify the proposed statistical analyses of multiple enteric pathogen outcomes. We originally provided greater detail about this approach in the pre-specified data-analysis plan (<https://osf.io/e7pvk/>); we elected to continue limiting illustrative model specifications and comprehensive interpretation of model parameters to the separate analysis plans to maintain the accessibility of this more general protocol, but have now directed readers more explicitly to these resources in the publicly available data analysis plan (lines 442-444):

The modelling approach, including illustrative model specifications for both binary and continuous outcomes and interpretation of model parameters, is described in the pre-specified data analysis plan (<https://osf.io/e7pvk/>).

We also indicate that the data analysis plan provides additional information on choice of prior distributions (lines 468-470):

The mixed effects models will be specified as Bayesian hierarchical models with regularizing hyperpriors (see data analysis plan for discussion of prior distributions) and sampled using Markov chain Monte Carlo (MCMC) approaches [50,93,96].

and on the censoring-based approach for zero-inflated, semi-continuous data on pathogen quantities (lines 480-484):

Differences in average gene copy density (scaled by pathogen-specific sample standard deviation to facilitate comparison across pathogens) will be estimated as the measure of effect using linear Bayesian hierarchical models for semicontinuous enteric pathogen quantity outcomes, with non-detects considered true zeros and censoring used to create a zero class (as in Tobit regression; refer to data analysis plan for implementation details) [96,99].

We have also greatly expanded the description of the proposed statistical approach in the manuscript. We do not propose a multivariate analysis in the sense of estimating a joint distribution of all the pathogen outcomes, and have taken care not to suggest as such in the revised manuscript. Instead, we propose treating the different pathogens as repeated measures of each child, with each specific pathogen representing a realization of the effect of the intervention on enteric pathogens generally. In practice this means constructing a “long” dataset in which the response vector holds the observed value for each pathogen for each child on separate rows, with the other child-specific covariates (e.g., age, sex) repeated for each pathogen. We will fit a multilevel model with a pathogen-varying intercept as well as pathogen-varying slopes for both the effect of the intervention and for all covariates, all of which will be modeled as arising from a multivariate normal distribution with a mean vector corresponding to the pooled average effect across all pathogens for each variable and with a covariance matrix with diagonal elements corresponding to the variance of each covariate effect across pathogens (with larger variances indicating greater variability between pathogens) and the off-diagonal elements corresponding to the dependence between effects (such as the relationship

between baseline prevalence, given by the pathogen-specific intercept, and the pathogen-specific treatment effect). This is essentially comparable to an individual participant meta-analysis in which each pathogen is treated as a separate study and the children in each study happen to be the same. We also allow the intercept to vary by child to account for the multiple pathogens measured in each child stool and by compound to account for the clustering of multiple children in each compound (the level at which the intervention was delivered). We will implement a Bayesian hierarchical framework using weakly informative priors to regularize the estimates by inducing mild shrinkage, which will both aid in stabilizing the estimates obtained by MCMC and help tame the estimated magnitude and better represent the precision of the effect estimates, which otherwise may be susceptible to so-called “Type-M” errors, in which the parameter estimates that manage to clear an arbitrary (if conventional) statistical significance threshold for a given realization are likely to be substantially exaggerated in magnitude relative to the true effect size. The proposed models will have large numbers of nominal parameters, but the use of pathogen-varying slopes for adaptive shrinkage supplemented with regularizing hyperpriors is expected to reduce the effective number of parameters to meaningfully fewer than sum of the parameters that would be estimated when fitting separate models for each pathogen, which could be viewed as a special case of our proposed model in which the group-level variance of each pathogen-specific effect is fixed at infinity (disallowing partial pooling of information between pathogens) rather than estimated from the data. We have expanded the description of the statistical analysis of multiple pathogens and its rationale accordingly (lines 440-466):

We will use mixed effects models with varying slopes and intercepts to simultaneously estimate individual treatment effects for each pathogen and the weighted-average intervention effect across all pathogens included in the model. The modelling approach, including illustrative model specifications for both binary and continuous outcomes and interpretation of model parameters, is described in the pre-specified data analysis plan (<https://osf.io/e7pvk/>). Briefly, the observed value (detection or gene copy density) of each pathogen for every child will be included as a separate response in the model design matrix and the intercept, intervention effect slope, and the slopes of other covariates will all be allowed to vary by pathogen. The intercept will also vary by child and compound to account for repeated measures of multiple pathogens per child and multiple children per compound. Each set of pathogen-varying effects (e.g., the pathogen-specific intervention effect slopes) will be structured as arising from a population of parameters with shared mean and variance [93,94], where the population-level mean corresponds to the weighted-average expected effect across all pathogens and the population-level variance indicates the extent to which the effect may differ by pathogen. We will also estimate covariances between the sets of pathogen-varying effects to account for dependencies, for example if the effect of the intervention on a specific pathogen is greater when the background prevalence of that pathogen (represented by the pathogen-specific intercept) is also higher. By partially pooling information between pathogens, this approach provides adaptive shrinkage of the individual treatment effect estimates for each pathogen as well as an estimate of the generalized effect across pathogens [50,93]. Such partial pooling of effect estimates helps control the false discovery rate for individual outcomes, mitigating the need for post hoc multiple comparison adjustments [69,70].

(1c)

Finally, it would be helpful to understand why a weighted average effect across 22 pathogens would be more interpretable than pathogen-specific estimates, or class-specific estimates (bacteria, protozoa, helminths, viruses). For example, what if there are 20 pathogens with clearly null effects and 2 pathogens with moderate reductions that appear to be real (as in the original analysis)? In this hypothetical scenario, wouldn't a weighted average effect across the 22 show no effect? Would the interpretation at the end of the day somehow be different than if the trial simply reported 22 separate effects? If the decision to estimate a weighted average effect is driven by statistical power, I would expect that 22 separate estimates with a reasonable false discovery rate correction for multiple

comparisons would provide nearly the same power and would be easier to understand/replicate and more interpretable.

We sincerely appreciate this feedback, which has helped us clarify our thinking on this matter. While we still believe there is demand for a single value to summarize the overall findings of a study (for reasons good and ill), and that our proposed pooled effect estimate—though still limited—is preferable to a composite outcome construct, we agree that this summary estimate would, in practice, typically be interpreted alongside the separate treatment effect estimates on individual pathogens. We have therefore revised the primary outcome to encompass both the pathogen-specific treatment effects and the weighted average effect across all 22 pathogens (lines 356-358):

Our primary outcome is the prevalence of a pre-specified subset of 22 enteric pathogens, including *Shigella* spp. and *T. trichiura*, the two pathogens most impacted at 24 months among children born after the intervention was delivered [34].

and lines 363-370:

Rather, we will estimate the effect of the intervention on the prevalence of each pathogen individually, along with a pooled estimate of the intervention effect across the outcome set to serve as a summary of intervention's impact on pathogen prevalence that is directly comparable to the individual effect estimates [69,70]

Corresponding language in the abstract and elsewhere has also been revised accordingly.

Our motivation for employing a single model encompassing multiple pathogen outcomes was driven less by increased power—welcome as it would be, and the potential for which we are actively investigating through simulations—and more by the opportunity to pool information about the treatment effect across pathogens to adaptively regularize the magnitude of treatment effect estimates for individual pathogens while also producing an interpretable summary of the overall effect of the intervention on a generic pathogen as a byproduct, all within a consistent framework. However, we would anticipate separate models for each pathogen with reasonable control for false discovery rate, as suggested by the reviewer, to yield broadly similar conclusions (if perhaps more exaggerated point estimates). We will therefore conduct sensitivity analyses with separate treatment effect estimates for each pathogen and post-hoc control for false discovery rate (lines 464-467):

As a sensitivity analysis, we will also fit separate models for each pathogen, applying the Benjamini-Hochberg procedure to control the false discovery rate [95].

and likewise for pathogen quantity outcomes (lines 484-489):

As a sensitivity analysis, we will also implement the two-stage parametric g-formula approach of Rogawski McQuade et al. to estimate differences in average quantity separately for each pathogen, applying the Benjamini-Hochberg procedure to account for multiple comparisons [11].

(2)

Page 22, lines 54-56

“Analyses will be repeated with missing covariate data imputed by multivariate imputation using chained equations (MICE) as a sensitivity analysis [34, 99].”

This relates to my previous comment about vague descriptions in the statistical analysis, but please explain how the 22-outcome, multivariate model for the primary analysis would then be incorporated

into a MICE routine for multiple imputation. It's difficult for me to understand how this would work, and the citations provided do not illustrate how it would be done.

We have expanded the description of the imputation procedure for both enteric pathogen outcomes and the secondary growth and caregiver-reported outcomes. Because we are conducting Bayesian inference using MCMC for the multiple enteric pathogen outcomes, we will fit these models to each of the imputed datasets and mix the resulting draws from the posterior distributions as suggested by Zhou and Reiter (2010; <https://doi.org/10.1198/tast.2010.09109>) and demonstrated in a vignette for the R package brms (https://paul-buerkner.github.io/brms/articles/brms_missings.html), which we have been using to perform MCMC for comparable models on similar TAC datasets. The updated text reads (lines 547-557):

Analyses will be repeated with missing data imputed by multiple imputation using chained equations (MICE) as a sensitivity analysis [34,105]. In addition to covariates, caregiver-reported outcomes and continuous growth outcomes (HAZ, WAZ, and WHZ) will be used for imputation; derived binary growth statuses (stunting, underweight, and wasting) will be re-calculated for each observation after conducting imputation procedures. Missing pathogen outcomes will not be imputed, nor will pathogens be used to inform imputation of other variables. However, datasets containing imputed covariates will be joined to observed pathogen outcomes by unique child identifier for sensitivity analyses; the Bayesian hierarchical models will be fit to each imputed dataset and draws from the posterior predictive distributions for parameters and contrasts of interest will be concatenated to obtain pooled estimates across the multiple imputed datasets [106].

(3)

Page 5, Lines 49-54. "...in particular, the presence of the intervention itself may have influenced the desirability of the intervention sites and thus the socioeconomic status of their residents, which may be associated with increased pathogen exposures" (also page 26, lines 6-41)

Please confirm the sentence should have "increased pathogen exposures" rather than "reduced pathogen exposures." Intuitively, I would have thought that the intervention would have potentially increased the desirability of the intervention sites, increased the socioeconomic status of residents, and in turn reduced pathogen exposure. If it is the other way around, it was not intuitive to me. Either way, perhaps it would help to connect the dots a bit more for readers and provide the rationale.

Thank you for identifying this oversight. We indeed expect increased SES to be associated with reduced pathogen exposures and have revised the text accordingly (lines 78-8):

... the presence of the intervention itself may have influenced the desirability of the intervention sites and thus the socioeconomic status of their residents, which may be associated with reduced pathogen exposures.

(4)

On the relative scale, the analysis proposes to use an odds ratio as a measure of effect to compare the weighted average prevalence of infection. Since infection prevalence will be high for many pathogens, the odds ratio will not approximate the prevalence ratio and might be exaggerated in the extreme. For example, if prevalence in the two groups were 80% and 85%, then the prevalence ratio is $0.80/0.85 = 0.94$ (a 6% relative reduction), but the odds ratio is $(0.8/0.2)/(0.85/0.15) = 0.71$ (a 29% relative reduction) — quite a difference! Since the authors propose to estimate prevalence differences using marginal predictions, why not do something similar for a more interpretable prevalence ratio on the relative scale? Furthermore, the authors propose estimating a prevalence ratio for growth-related binary outcomes. It would seem like a consistent analysis approach and

parameter of interest might be valuable across similar outcomes in the same study, or at least some justification for why they should differ.

This is an excellent suggestion and we fully agree that prevalence ratio estimates derived from marginal predictions would be more interpretable as measures of effect on the relative scale than conditional prevalence odds ratios. We also do not have a strong justification for favoring relative effect measures over absolute effects, and have revised the text to afford equal standing to both marginal prevalence ratios and marginal prevalence differences for the primary outcome analysis (lines 470-480):

For binary enteric pathogen detection outcomes, the canonical logistic link function will be used for computational stability and a Bayesian parametric g-formula algorithm will be applied to estimate marginal prevalence ratios and prevalence differences from the posterior predictive distribution [97]. Posterior predicted probabilities will be sampled for each pathogen assuming all participants received the intervention and again assuming none received the intervention, leaving all other covariate values unchanged, to obtain weighted averages over the distribution of confounders in the sample population, as in marginal standardization [98]. The posterior predicted prevalence ratio and prevalence difference distributions for each pathogen will be approximated by the ratio and difference, respectively, of the posterior predicted probability draws under the all-treated and none-treated scenarios.

(5)

Page 21, lines 42-46

“Prevalence differences (PD) between children in intervention and control compounds will also be estimated from the posterior predictive distribution at representative values of other model covariates [92].”

Why a predictive distribution at fixed values rather than simply use marginal standardization? There are substantive differences in interpretation, and in general, marginal predictions are often easier to interpret: <https://pubmed.ncbi.nlm.nih.gov/24603316/>

Thank you for bringing this to our attention. We agree that marginal standardization is a preferable approach that will yield more interpretable estimates. Because we are using MCMC to fit Bayesian hierarchical models, we have opted to implement the Bayesian parametric g-formula to estimate marginal prevalence ratios and prevalence differences using draws from the posterior predictive distributions already available to us (lines 470-480):

For binary enteric pathogen detection outcomes, the canonical logistic link function will be used for computational stability and a Bayesian parametric g-formula algorithm will be applied to estimate marginal prevalence ratios and differences from the posterior predictive distribution [97]. Posterior predicted probabilities will be sampled for each pathogen assuming all participants received the intervention and again assuming none received the intervention, leaving all other covariate values unchanged, to obtain weighted averages over the distribution of confounders in the sample population, as in marginal standardization [98]. The posterior predicted prevalence ratio and prevalence difference distributions for each pathogen will be approximated by the ratio and difference, respectively, of the posterior predicted probability draws under the all-treated and none-treated scenarios.

Reviewer 2:

Dr. Colette Mmapenya Khabo-Mmekoa, Tshwane University of Technology

(1)

Can the authors please specify when this study began? As stated on page 16, first paragraph, last sentence, participant enrollment did not begin until March 2022. When did the onsite sanitation interventions take place?

Thank you for the opportunity to clarify the timing of the intervention, which was originally implemented in 2015 and 2016. The baseline survey for the original study was conducted at this time, shortly before the intervention facilities were completed and opened for use by the receiving compounds. However, the same implementing partner constructed similar sanitation facilities in the same neighborhood prior to 2015 and continued to deliver this intervention in 2017 and onwards. As a cross-sectional follow-up study of an existing sanitation intervention, we are enrolling any compound we identify that have received the sanitation intervention (built by WSUP to the same specifications) at any time, provided that the intervention has been in place for a minimum of 5 years at the time of enrollment. While we anticipate the majority of interventions enrolled are from the initial 2015-2016 intervention period, facilities constructed up to a few years before or after will also be eligible, if identified. Under the description of the urban sanitation intervention we write (lines 197-200):

The nongovernmental organization (NGO) Water and Sanitation for the Urban Poor (WSUP) implemented a sanitation intervention in 2015-2016 at approximately 300 compounds in informal urban neighborhoods of Maputo under a larger program led by the Water and Sanitation Programme (WSP) of the World Bank [47].

And we have added the following clarification to the end of this section (lines 210-212):

WSUP also constructed facilities to the same specifications at other compounds in the neighborhoods prior to 2015 and continued to deliver these intervention designs in the years following 2016.

Additionally, in the description of the design of the current long-term follow-up study we explain (lines 244-250):

Due to substantial population turnover [34], both intervention and control compounds are being identified using previously collected geolocation data and their locations confirmed by study staff during site identification visits. Any compounds identified during the course of these mapping efforts that have received the sanitation intervention at least five years prior (built by WSUP to the same specifications) or match the original enrollment criteria for MapSan control compounds are also invited to participate, irrespective of previous participation in the MapSan trial.

(2)

Authors must take note of the citation of the references. All references are recorded accordingly up to page 22. Reference 100 is listed but not cited within the text. This may be a need for adjustment for the remaining references.

The reference list has been updated with additional citations in response to reviewer feedback. Accordingly, we have thoroughly checked all the references in the revised text and ensured each is cited appropriately.

Reviewer 3:

Dr. Tim Foster, University of Technology Sydney

(1)

My only suggestion is in relation to the 'public domain' environmental sampling, where a bit more detail might be helpful. While the protocol articulates the approach for the environmental sampling at control and intervention compounds, I could not find information on the proposed sample size for the public domain sampling, the expected distribution across public domain sample types (wastewater, surface water, soil etc) or how the public domain locations would be selected.

Thank you for the suggestion. We have expanded the description of the public domain environmental sampling to include this additional information (lines 317-331):

We are also collecting environmental samples from the public domain to conduct environmental surveillance of pathogens circulating in the community [56,60]; such samples are not linked to specific individuals in any way. Sample matrices are informed by the compartments on Maputo's Excreta Flow Diagram [61], including wastewater, surface and open drain water, soils in the vicinity of solid and fecal waste disposal locations, and wastewater effluent from hospitals treating COVID-19 patients. We identify sample locations using satellite imagery in consultation with technicians responsible for sanitation and drainage in Maputo. Samples are collected with written permission from the municipal government on a weekly basis for matrices with a single available sampling location (e.g., wastewater treatment plant influent) and twice at all other locations (once in each the rainy and dry seasons) to prioritize the geographic distribution of sampling locations across Maputo city. We aim to collect approximately 100 samples of solids (soil, fecal sludge), 100 large-volume liquid samples (wastewater, surface water, and open drains) [62], and 100 passive samples of liquids [60].

VERSION 2 – REVIEW

REVIEWER	Arnold, Ben UCSF, F.I. Proctor Foundation
REVIEW RETURNED	27-Apr-2023
GENERAL COMMENTS	I commend the authors on an exceptional job revising this protocol. The statistical methods are quite novel and much more clearly described in the revision. Thank you for the clarification and additional detail, which greatly enhances the exposition. I wish the team good luck with this important follow-up study of a landmark trial in the WASH sector.

VERSION 2 – AUTHOR RESPONSE